# NKG2D Polymorphism in Melanoma Patients from Southeastern Spain

**DOI:** 10.3390/cancers11040438

**Published:** 2019-03-28

**Authors:** Lourdes Gimeno, Helios Martínez-Banaclocha, M. Victoria Bernardo, José Miguel Bolarin, Luis Marín, Ruth López-Hernández, M. Rocío López-Alvarez, M. Rosa Moya-Quiles, Manuel Muro, José Francisco Frias-Iniesta, Jorge Martínez-Escribano, M. Rocío Alvarez-López, Alfredo Minguela, José Antonio Campillo

**Affiliations:** 1Immunology Service, Hospital Clínico Universitario Virgen de la Arrixaca (HCUVA), Instituto Murciano de Investigación Biomédica (IMIB), 30120 Murcia, Spain; lourdes.gimeno@carm.es (L.G.); bolarin5@hotmail.com (J.M.B.); rosa.moya2@carm.es (M.R.M.-Q.); manuel.muro@carm.es (M.M.); mdrocio.alvarez@gmail.com (M.R.A.-L.); josea.campillo@carm.es (J.A.C.); 2Inflammation and Experimental Surgery Unit, IMIB, 30120 Murcia, Spain; heliosmar@live.com; 3Department of Pharmacy, Faculty of Health Sciences, Universidad Católica de San Antonio de Murcia (UCAM), 30107 Murcia, Spain; vbernardo@ucam.edu; 4Immunology Section, Complejo Hospitalario Universitario de Albacete, 02006 Albacete, Spain; lamarin24@gmail.com; 5Immunology Section, Hospital Universitario de Gran Canaria Doctor Negrín, 35010 Las Palmas de Gran Canaria, Spain; rlophera@gobiernodecanarias.org; 6Center for Preventive Medicine, Animal Health Trust, Lanwades Park, Kentford, Newmarket, Suffolk CB8 7UU, UK; MARIA.LOPEZ@aht.org.uk; 7Dermatology Service, Hospital Clínico Universitario Virgen de la Arrixaca (HCUVA), Instituto Murciano de Investigación Biomédica (IMIB), 30120 Murcia, Spain; josef.frias@carm.es (J.F.F.-I.); jorge.mescribano@gmail.com (J.M.-E.)

**Keywords:** Melanoma, NK cell, NKG2D, NKG2A, gene polymorphism

## Abstract

Background: Natural killer (NK) and CD8+ T cells are involved in the immune response against melanoma. C-Type lectin-like NK cell receptors are located in the Natural Killer Complex (NKC) region 12p13.2-p12.3 and play a critical role in regulating the activity of NK and CD8+ T cells. An association between polymorphisms in the NKC region, including the *NKG2D* gene and *NKG2A* promoter, and the risk of cancer has been previously described. The aim of this study was to analyze the association of polymorphisms in the NKC region with cutaneous melanoma in patients from southeastern Spain. Methods: Seven single-nucleotide polymorphisms (SNPs) in the *NKG2D* gene (NKC3,4,7,9,10,11,12), and one SNP in the *NKG2A* promoter (NKC17) were genotyped by a TaqMan 5′ Nuclease Assay in 233 melanoma patients and 200 matched healthy controls. Results: A linkage disequilibrium analysis of the SNPs performed in the NKC region revealed two blocks of haplotypes (Hb-1 and Hb-2) with 14 and seven different haplotype subtypes, respectively. The third most frequent haplotype from the block Hb-2—NK3 (CAT haplotype)—was significantly more frequent on melanoma patients than on healthy controls (*p* = 0.00009, Pc = 0.0006). No further associations were found when NKC SNPs were considered independently. Conclusions: Our results suggest an association between *NKG2D* polymorphisms and the risk of cutaneous malignant melanoma.

## 1. Introduction

The incidence of cutaneous malignant melanoma is increasing worldwide and, although it only accounts for 5% of cutaneous malignancies, this tumor is responsible for approximately 75% of all deaths caused by skin cancer [1,2,3]. Malignant melanoma is one of the most immunogenic tumors [4,5]; nevertheless, it can escape the host’s immune surveillance, especially when a state of tolerance is induced [6,7,8]. 

Natural killer (NK) cells are an important component of the innate immune response. NK cells are able to eliminate target cells without previous sensitization [4,5,6,7,8]. Together with antigen-specific cytolytic CD8+ T cells, NK cells are key components in the response against infected and transformed cells through potent cytolytic activity [4,5,9,10,11]. The effector functions of NK and CD8+ T cells are regulated by a balance of signals delivered by activating and inhibitory receptors that interact with their cognate ligands expressed on infected or transformed cells [12,13,14], including malignant melanocytes [9,15]. In this context, it should be noted that the NK cell-mediated killing of melanoma cells is controlled by multiple activating receptors/ligands [10,16,17,18], among which the NKG2D receptor/NKG2D ligand interaction plays an important role and induces one of the most important signals leading to the lysis of melanoma tumor cells [9,19].

NKG2D (KLRK1) is a C-type lectin-like type II integral membrane molecule involved in immunosurveillance, with costimulatory and activating functions [20,21,22]. NKG2D is expressed on the surface of a variety of immune cells including NK cells, CD8+ αβ and γδ T cells, and activated macrophages. In NK cells, NKG2D associates with DAP10-phosphoinositide 3-kinase signaling pathway to trigger cell-mediated cytotoxicity. NKG2D ligands are the major histocompatibility complex (MHC) class I chain-related proteins A and B (MICA/B) and UL16 binding proteins 1 (ULBP1) and 2 (ULBP2) [20,21,22,23,24]. MICA and MICB are not usually expressed on normal cells but they are found at low levels on intestinal epithelial cells and are induced by cellular stress, typically on tumor or virus-infected cells [25,26]. 

NKG2A is a C-type lectin-like receptor expressed in the form of a NKG2A/CD94 heterodimer on NK and T cells. NKG2A interacts with the nonclassical Human Leukocyte Antigen (HLA)-E molecules that present leader peptides from other HLA class-I alleles [17,19], so that NKG2A^+^ NK cells can survey autologous cells for overall HLA levels. NKG2A/HLA-E interaction delivers inhibitory signals playing a decisive role in NK cell education mostly of the less-differentiated CD56^bright^ NK cell subset. NK cells become educated and fully competent during a process known as “licensing”, where NKG2A and/or inhibitory killer cell immunoglobulin-like receptors (iKIR) interact with their cognate HLA-I ligands [27].

The gene encoding NKG2D is located on human chromosome 12, within the Natural Killer Complex (NKC) region (12p13.2-p12.3), where limited polymorphism has been described [28]. NKG2D polymorphism has been previously associated with a variety of pathologies, including viral [29] and autoimmune diseases [30], as well as with immune surveillance against cancer [19,31,32,33]. Even though NKG2D is one of the best characterized primary activating receptors on NK cells, several factors, including tumor microenvironment and genetic variation, impact the efficacy of the NKG2D-mediated antitumor response and, consequently, on the patient´s clinical outcome. In fact, specific haplotypes in the NKC region, defined by several single-nucleotide polymorphisms (SNPs) located in the *NKG2D* and *NKG2A* genes, have been associated with variations in NK cell cytotoxic activity as well as with the overall cancer risk in a 2006 study of a Japanese population [28]. The aim of our study was to investigate the influence of the previously described polymorphisms in the NKC region [28], which included SNPS of *NKG2D* and *NKG2A* genes, on the risk of developing cutaneous melanoma in a series of patients from Murcia, a Mediterranean Spanish population. A higher risk of cutaneous malignant melanoma in individuals bearing the NK-3 Hb-2 haplotype was found. Additionally, a tendency towards a protective effect of the NK-2 Hb-2 haplotype (previously associated with a higher cytotoxic activity of NK and TCD8^+^ cells) was also revealed.

## 2. Results

### 2.1. Clinical and Histological Characteristics of Melanoma Patients

As summarized in Table 1, 162 patients were histologically diagnosed with superficial spreading melanoma (SSM), 29 with nodular melanoma (NM), 16 with lentigo maligna melanoma (LMM), 13 with acral lentiginous melanoma (ALM), 11 with melanoma “in situ”, one with desmoplastic melanoma (DM), and one with Spitzoid melanoma (SM). Patients were classified according to tumor thickness (≤1.0 mm vs. >1.0 mm), the presence or absence of ulceration on the primary lesion, and the sentinel lymph node (SLN) status (metastatic vs. nonmetastatic). Data about melanoma tumor thickness and ulceration were available for 232 and 217 out of 233 patients, respectively, and SLN biopsy was performed on 141 patients, of whom 16% showed metastatic SLNs (Table 1).

### 2.2. Frequency of SNPs in the NKC Region

The frequencies of seven SNPs in the *NKG2D* gene (NKC3, 4, 7, 9, 10, 11, and 12) and one SNP in the *NKG2A* promoter (NKC17) were analyzed in both the patients and the control group. Frequencies of alleles and genotypes are summarized in Table 2. A decreased frequency of NKC7 A-allele in patients compared to controls (50% vs. 60%; *p* = 0.042/Pc = 0.34, OR = 0.66) was initially found. However, this association was not statistically significant after Bonferroni correction. We also observed a reduction in the frequency of NKC7 A-allele in patients with SSM compared to the control group (49% versus 60%; *p* = 0.03/Pc = 0.24, OR = 0.6), but this association was not found after *p*-correction. No differences between the frequencies of the remaining SNPs of melanoma patients and healthy controls were observed.

We also studied NKC genotypes. The frequency of the NKC7TT genotype was increased in both the total group of melanoma patients and the SSM patients when compared to healthy controls (51% vs. 40%; *p* = 0.042, Pc = 1.01, OR = 1.51; *p* = 0.03, Pc = 0.72, OR = 1.5, respectively). By contrast, the frequency of the AG genotype in the NKC9 SNP was lower in melanoma patients and SSM patients than in the control group (25% and 23% vs. 35%; *p* = 0.026, Pc = 0.62, OR = 0.62; *p* = 0.021, Pc = 0.50, OR = 0.6, respectively). None of these observations remained significantly different after Bonferroni correction.

We next analyzed the frequency of the NKC SNPs alleles and genotypes according to the clinical characteristics of melanoma patients at diagnosis (Table 3). The frequencies of the G allele in NKC9, the T allele in NKC11 and the C allele in NKC12 were increased in melanoma patients with ulceration when compared to patients without ulceration (98%, 68%, and 73% vs. 86%, 51%, and 56%; *p* = 0.033, Pc = 0.26, OR = 7; *p* = 0.043, Pc = 0.34, OR = 2; *p* = 0.041, Pc = 0.33, OR = 2.1, respectively). By contrast, a decreased frequency of the G allele in NKC9 was observed in melanoma patients with SLN metastasis when compared to the control group (77% vs. 92%; *p* = 0.042, Pc = 0.34, OR = 0.3). Besides, decreased frequencies of the C allele in NKC4 and the T allele in NKC10 were found in metastatic SLN compared to no metastatic SLN (86% and 86% vs. 98% and 98%; *p* = 0.049, Pc = 0.39, OR = 1.32; *p* = 0.049, Pc = 0.39, OR = 0.16, respectively). However, these differences in the frequency of the analyzed NKC SNPs were not statistically significant after *p*-value correction. 

The study of genotypes showed a decreased frequency of the AA genotype in NKC9, the CC genotype in NKC11, and the GG genotype in NKC12 in patients with ulceration compared to those without ulceration (2%, 32%, and 27% vs. 14%, 49%, and 45%; *p* = 0.033, Pc = 0.79, OR = 0.1; *p* = 0.043, Pc = 1.03, OR = 0.48; *p* = 0.04, Pc = 0.96, OR = 0.46, respectively). The frequency of the AA genotype in NKC10 was increased in patients with SLN metastasis compared to the control group (23% vs. 8%; *p* = 0.042, Pc = 1.01, OR = 3.4). Similarly, the frequency of AA genotype in NKC9 was also increased in patients with SLN metastasis compared to those that did not show metastasis (14% vs. 2%; *p* = 0.049, Pc = 1.18, OR = 6.1). No statistically significant differences were observed after Bonferroni correction.

### 2.3. Identification of NKC Haplotypes

Linkage disequilibrium (Figure 1) showed that the SNPs NKC3, NKC7, NKC11, and NKC12 were closely linked to each other, with r^2^ values >0.7. On the other hand, the SNPs NKC4, NKC9, and NKC10 were also closely linked (r^2^ > 0.7). These results allowed us to identify two main haplotype blocks named Hb-1 (including NKC3, NKC7, NKC11, and NKC12 SNPs) and Hb-2 (including NKC4, NKC9, and NKC10 SNPs) (Figure 1A). Moreover, we found that the SNP NKC17 on *NKG2A* promoter region was not linked to the rest of the SNPs analyzed. Finally, we studied Hb-1 and Hb-2blocks in control and patient groups and found fourteen different haplotypes for Hb-1 and seven for Hb-2 (see Figure 1B).

### 2.4. NKC Haplotype Frequencies

We estimated the haplotype frequencies in melanoma patients and controls (see Table 4). This analysis showed that NK-1 and NK-2 haplotypes from the Hb-1 block were the most frequent haplotypes in both the melanoma (freq = 0.619 and 0.276, respectively) and the control (freq = 0.587 and 0.329, respectively) groups, showing similar frequencies in both groups. However, the frequency of the NK-3 haplotype from the Hb-1 block was higher in melanoma patients than in healthy controls (4.8% vs. 1.5%; *p* = 0.007, Pc = 0.098). In contrast, the frequency of the NK-10 haplotype from the Hb-1 block was lower in melanoma patients than in controls. These differences did not remain statistically significant after *p*-correction for multiple testing (0% vs. 1%; *p* = 0.03, Pc = 0.42).

We also studied the frequency of the NKG2D Hb-2 block haplotypes in patients and controls (Table 5). Interestingly, melanoma patients showed a higher frequency of the NK-3 haplotype from the Hb-2 block when compared to the control group (0.039 vs. 0.0, *p* = 0.00009, Pc = 0.0006). No significant differences in the frequency of the NK-3 haplotype from the Hb-2 block were found between SSM and NM histological subtypes or tumor thickness >1mm, although the frequency found in patients with ulceration (0.000, *p* < 0.05, Pc = 0.28) was lower than in patients without ulceration (0.046) (Table 6).

### 2.5. NKC Haplotypes and Melanoma Risk

Finally, we estimated the risk of melanoma for the two most common haplotypes (NK-1 and NK-2) from Hb-1 and Hb-2 blocks in the most widespread histological subtypes of melanoma (SSM and NM, Table 7). No association between the frequencies of the different haplotypes and the risk of developing the most common types of melanoma was found.

## 3. Discussion

In the present study we have researched the polymorphisms in the NKC region in a series of patients with cutaneous malignant melanoma from Southeastern Spain. We have studied eight SNPs: seven located on the coding region of *NKG2D* gene and one in the promoter region of *NKG2A*. Some of these SNPs have been previously associated with susceptibility to infectious diseases, autoimmunity, or cancer [29,30,31,34,35,36,37]. Our results show no association between the risk of developing cutaneous melanoma and the NKC SNPs when considered individually. However, a higher frequency of the NK-3 haplotype from Hb-2 block was observed in melanoma patients compared to the control group, suggesting that *NKG2D* polymorphisms may influence the onset of cutaneous melanoma. Although no other studies to which we can compare our results have been performed on patients with melanoma, the CC genotype of the NKC3 SNP has been associated with susceptibility to breast cancer in Iranian patients [35]. The 72Thr variant of NKC4 SNP (rs2255336) showed a protective effect against cervical cancer in Polish patients [36], and the rs11053781 and rs2617167 SNPs of *NKG2D* increased the risk of developing cholangiocarcinoma in a cohort of European patients with primary sclerosing cholangitis [31]. However, no association was observed between NKC polymorphisms and cholangiocarcinoma in patients from the USA [37]. All these results together suggest that the polymorphism of the NKC region, especially in the NKG2D gene, would be related to the susceptibility to different types of cancer. However, important differences in the types of cancer, the ethnic groups, and the genetic variants analyzed suggest the need for standardized studies in homogeneous and broader series to draw more definitive conclusions.

The linkage disequilibrium analysis amongst the SNPs studied on the NKC region allowed for the identification of two haplotype blocks: haplotype block 1 (Hb-1), consisting of NKC3, NKC7, NKC11, and NKC12 SNPs and haplotype block 2 (Hb-2), consisting of NKC4, NKC9, and NKC10 SNPs. These haplotype blocks differed from those described in the Japanese population [28], where the NKC17 SNP was associated with the NKC3, NKC7, NKC11, and NKC12 SNPs of the Hb-1 block. In the present study no association was found between the NKC17 and the rest of the SNPs analyzed. Therefore, the Hb-1 block haplotypes described in this manuscript differ from those described previously in the Japanese population [28], probably due to the existence of ethnic variations in the composition of these haplotypes or the presence of a novel finding. Nevertheless, the most frequent haplotypes of the Hb-2 block in our population correspond with those described in the Japanese population. 

Although the analysis of the two major haplotypes of the Hb-1 and Hb-2 blocks showed no association with the risk of developing melanoma, the frequencies of the NK-2 haplotypes from Hb-1 and Hb-2 blocks and the NK-1/NK-2 and NK-2/NK-2 genotypes for the Hb-2 block were decreased in melanoma patients, but these associations were not statistically significant. These results suggest that the presence of the NK-2 Hb-2 haplotype, previously related to a high cytotoxic activity of NK and CD8+ T cells [28], could be associated with a protective effect against the development of cutaneous melanoma. However, studies in larger series and functional assays would be necessary to demonstrate higher NK cell cytotoxic activity in our population associated with the NK-2 Hb-2 haplotype block. 

In the same way, other studies have described an association between the higher cytotoxicity of HNK1 haplotype (equivalent to NK-2 Hb-1 haplotype in this study) and a better clinical outcome in standard risk hematologic malignancies after bone marrow transplant of unrelated HLA compatible donors [38], as well as a reduced risk of developing colorectal cancer [39,40]. However, in the present study, no associations were observed between the two most frequent haplotypes from the Hb-1 and Hb-2 blocks and the susceptibility to cutaneous melanoma. Conversely, an association between the NK-3 Hb-2 haplotype and an increased risk of developing cutaneous melanoma was found in our population. But, due to the low frequency of the NK-3 Hb-2 haplotype, studies in a larger series are required to confirm this association. In addition, it would also be interesting to perform functional studies to determine if this haplotype is associated with a reduced cytotoxic activity of NK and CD8+ T cells, which would explain the observed susceptibility to cutaneous melanoma. 

## 4. Materials and Methods

### 4.1. Subjects

A total of 233 Caucasian patients (112 men and 121 women, 53 ± 15 years of age) histologically diagnosed with cutaneous malignant melanoma in the Dermatology Service of Virgen de la Arrixaca University Hospital in Murcia, Spain were included in this study between 1996 and 2013. Patient classification was made following the recommendations of the American Joint Committee on Cancer (AJCC) [41,42]. Exclusion criteria were a history of another malignant, autoimmune, inflammatory, or infectious chronic disease, or immunodeficiency. Additionally, a series of 200 sex- and age-matched Caucasian healthy individuals of the same ethnic origin were studied (92 men and 108 women, 51 ± 17 years). All individuals included in the present study were unrelated and randomly selected. The study was approved by the Research Ethics Committee and the Institutional Review Board (IRB-00005712). Written informed consent was obtained from all patients and controls in accordance with the Declaration of Helsinki.

### 4.2. Sample Collection and DNA preparation

Blood samples from patients and healthy individuals were obtained by venipuncture in anticoagulant Vacutainer tubes (Becton Dickinson, Mountain View CA, USA). Genomic DNA was extracted from peripheral leucocytes by using the QIAamp DNA Blood Mini Kit (QIAGEN, GmbH, Germany) according to the manufacturer’s instructions.

### 4.3. NKC Region Genotyping

Eight SNPs of the NKC region closely associated with NK cell cytotoxic activity according to Hayashi et al. [28] were genotyped: seven located in the *NKG2D* gene and one in the promoter region of *NKG2A* (see Table 8). Following the criteria described by Hayashi et al., these SNPs were selected among the 1300 SNPs registered for this region in the Celera Genomic database and NCBI database. Briefly, selection was based on (1) allele frequency >10% in Caucasian and Japanese populations and (2) strong association with NK cell cytotoxic activity (*p*-values < 0.001) [28]. Genotyping was made using TaqMan-Allelic discrimination methods in a 7500-fast real-time polymerase chain reaction system (Applied Biosystems (AB), CA). Predesigned TaqMan assays for the selected SNPs were acquired from AB. The NKC10 SNP was a custom design made by TaqMan assay design tool on the AB website (NKC10 Forward primer: GGAGAAAAGGACATGCCCTCATAT; NKC10 Reverse primer: GTCTCTAAAGGGATGCAAAATGATAATAAAATGT). Results were analyzed using allelic discrimination software (AB). 

### 4.4. NKC Region Haplotype Analysis

Linkage disequilibrium (LD) analysis was carried out considering the eight NKC SNPs typed in order to identify the haplotype blocks previously described in the Japanese population [28]. All patients were included in the LD analysis (*N* = 233). Since the gametic phase was unknown, haplotype frequencies were estimated by an expectation–maximization algorithm leading to maximum likelihood estimates of gene frequencies, using the Arlequin software package version 3.0 [43]. Hardy–Weinberg equilibrium was tested applying a modified hidden Markov chain with a 100,000-step approach, as implemented in the Arlequin program [43]. LD analysis was made with the Haploview program v 4.0 [44].

### 4.5. Statistical Analysis

Demographic and experimental data were collected in a database (Microsoft Access 11.0; Microsoft corporation, Seattle, WA), and statistical analysis was performed using SPSS 15.0 software (SPSS Inc., Chicago, IL, USA). Chi-squared tests and two-tailed unpaired *t*-tests were used to detect differences regarding sex and age, respectively. NKC allele frequencies were estimated by direct counting and the percentage of positive individuals for a certain allele represented. 

Associations between polymorphisms of the NKC region and melanoma were established by comparing the frequencies of each NKC SNP allele in patients and healthy controls, using Chi-square and Fisher’s exact tests. Odds ratio (OR) and its 95% confidence interval (CI) were calculated to estimate any relative risk. *p*-value < 0.05 was considered significant. Corrected *p*-values (Pc) were obtained by multiplying the *p*-value by the number of alleles tested according to Bonferroni correction [45].

## 5. Conclusions

Although our results show no significant association between individual NKC SNPs and cutaneous melanoma in a Mediterranean population from the southeast of Spain, a higher risk in individuals bearing the NK-3 Hb-2 haplotype was found. Additionally, a tendency towards a protective effect of the NK-2 Hb-2 haplotype (previously associated with higher cytotoxic activity of NK and TCD8+ cells) was also revealed. Nonetheless, these results need to be confirmed in larger series of melanoma patients from different ethnic origins.

## Figures and Tables

**Figure 1 cancers-11-00438-f001:**
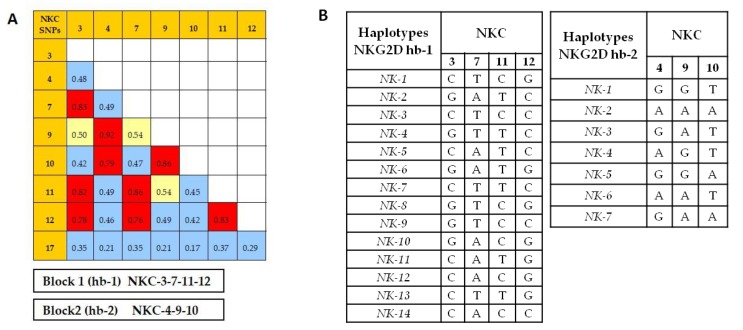
Linkage disequilibrium analysis between the eight NKC SNPs analyzed in melanoma patients in the Haploview software. (**A**) The r^2^ values of the association with each SNP are represented: r^2^ values > 0.7 (high grade of linkage disequilibrium) in red; r^2^ values between 0.5 and 0.7 in yellow; and r^2^ values < 0.5 in blue. (**B**) The two blocks of SNPs in close linkage are also represented; block Hb-1 including SNPs NKC3, NKC7, NKC11, and NKC12, and block Hb-2 including SNPs NKC4, NKC9, and NKC10.

**Table 1 cancers-11-00438-t001:** Clinical and demographic characteristics of melanoma patients and controls.

Demographic Data	Controls ^1^ (*N* = 200)	Melanoma Patients (*N* = 233)
Age (years, Mean ± SD)	51 ± 17	53 ± 15
Gender (Male)	92 (46%)	112 (48%)
Histological subtypes		
SSM		162 (69%)
NM		29 (12%)
LMM		16 (7%)
ALM		13 (6%)
Other ^2^		13 (6%)
Clinical characteristics		
Tumor thickness		
≤1 mm		128 (55%)
>1 mm		104 (45%)
Ulceration at primary lesion		
Yes		44 (20%)
No		173 (80%)
SLN metastasis		
Yes		22 (16%)
No		119 (84%)

^1^ 200 sex- and age-matched healthy Caucasian individuals of the same ethnic origin were included (92 men and 108 women). SSM: superficial spreading melanoma; NM: nodular melanoma; LMN: lentigo maligna melanoma; ALM: acral lentiginous melanoma; SLN: sentinel lymph node. ^2^ Other: Melanoma “in situ” (*n* = 11, 5%); Desmoplastic melanoma (*n* = 1, 0.5%); Spitzoid melanoma (*n* = 1, 0.5%).

**Table 2 cancers-11-00438-t002:** NKC allele and genotype frequencies in healthy controls and melanoma patients.

NK CRegion	Genotype	Controls (*N* = 200)	Melanoma Patients (*N* = 233)		
			Total	SSM	NM
NKC3	CC	83 (42%)	114 (49%)	81 (50%)	13 (45%)
	CG	86 (43%)	94 (40%)	62 (38%)	14 (48%)
	GG	31 (15%)	25 (11%)	19 (12%)	2 (7%)
	C-allele	169 (85%)	208 (89%)	143 (88%)	27(93%)
NKC4	GG	111 (56%)	138 (59%)	97 (60%)	15 (52%)
	AG	72 (36%)	85 (37%)	58 (36%)	13 (45%)
	AA	17 (8%)	10 (4%)	7 (4%)	1 (3%)
	G-allele	183(92%)	223 (96%)	155 (96%)	28 (97%)
NKC7	AA	28 (14%)	22 (9%)	17 (11%)	2 (7%)
	AT	92 (46%)	94 (40%)	62 (38%)	14 (48%)
	TT	80 (40%)	117 (51%) ^1^	83 (51%) ^2^	13 (45%)
	A-allele	120 (60%)	116 (49%) ^3^	79 (49%)^4^	16 (55%)
NKC9	AA	16 (8%)	27 (12%)	21 (13%)	3 (10%)
	AG	70 (35%)	58 (25%) ^5^	38 (23%) ^6^	9 (31%)
	GG	114 (57%)	148 (63%)	103 (64%)	17 (59%)
	G-allele	184 (92%)	206 (88%)	141 (87%)	26 (90%)
NKC10	AA	14 (7%)	9 (4%)	7 (4%)	1 (3%)
	AT	75 (38%)	76 (33%)	53 (33%)	10 (35%)
	TT	111 (55%)	148 (63%)	102 (63%)	18 (62%)
	T-allele	186 (93%)	155 (96%)	155 (96%)	28 (97%)
NKC11	CC	80 (40%)	109 (47%)	79 (49%)	12 (41%)
	CT	91 (46%)	99 (42%)	63 (39%)	15 (52%)
	TT	29 (14%)	25 (11%)	20 (12%)	2 (7%)
	T-allele	120 (60%)	124 (53%)	83 (51%)	17 (59%)
NKC12	CC	32 (16%)	35 (15%)	26 (16%)	3 (10%)
	CG	87 (44%)	100 (43%)	64 (40%)	17 (59%)
	GG	81 (40%)	98 (42%)	72 (44%)	9 (31%)
	C-allele	119 (60%)	135 (58%)	90 (56%)	20 (69%)
NKC17	CC	35 (17%)	43 (18%)	32 (20%)	7 (24%)
	CG	89 (45%)	112 (48%)	73 (45%)	14 (48%)
	GG	76 (38%)	78 (34%)	57 (35%)	8 (28%)
	C-allele	124 (62%)	155 (66%)	105 (65%)	21 (72%)

SSM: superficial spreading melanoma; NM: nodular melanoma. ^1,2^ Total melanoma patients or SSM vs. control (*p* = 0.042, Pc = 1.01, OR = 1.51; *p* = 0.026, Pc = 0.62, OR = 0.62, respectively). ^3^ Melanoma patients vs. control (*p* = 0.042, Pc = 0.34, OR = 0.66). ^4^ SSM vs. control (*p* = 0.03, Pc = 0.24, OR = 0.6). ^5,6^ Total melanoma patients or SSM vs. control (*p* = 0.03, Pc = 0.72, OR = 1.5; *p* = 0.021, Pc = 0.50, OR = 0.6). *p*-value was determined by two-sided Fisher’s exact test.

**Table 3 cancers-11-00438-t003:** NKC allele and genotype frequencies in healthy controls and melanoma patients according to clinical characteristics at diagnosis.

NKC Region	Genotype	Control *N* = 200	Melanoma Patients (*N* = 233)					
			Tumor Thickness		Ulceration		LN Metastasis	
			≤1.0 mm *N* = 128	>1.0 mm *N* = 104	No *N* = 173	Yes *N* = 44	No *N* = 119	Yes *N* = 22
NKC3	CC	83 (42%)	63 (49%)	51 (49%)	84 (49%)	20 (46%)	58 (49%)	11 (50%)
	CG	86 (43%)	50 (39%)	43 (41%)	70 (41%)	19 (43%)	50 (42%)	6 (27%)
	GG	31 (15%)	15 (12%)	10 (10%)	19 (11%)	5 (11%)	11 (9%)	5 (23%)
	C-allele	169 (85%)	113 (88%)	94 (90%)	154 (90%)	39 (89%)	108 (91%)	17 (77%)
NKC4	CC	111 (56%)	74 (58%)	63 (61%)	103 (60%)	22(50%)	73 (61%)	12 (55%)
	CT	72 (36%)	47 (37%)	38 (37%)	61(35%)	21 (48%)	43 (36%)	7 (32%)
	TT	17 (8%)	7 (6%)	3 (3%)	9(5%)	1 (2%)	3 (2%)	3 (14%)
	C-allele	183 (92%)	121 (95%)	101 (98%)	164 (95%)	43 (98%)	116 (97%)	19 (87%) ^1^
NKC7	AA	28 (14%)	14 (11%)	8 (8%)	16 (9%)	5 (11%)	9 (8%)	4 (18%)
	AT	92 (46%)	48 (38%)	45 (43%)	68 (39%)	21 (48%)	50 (42%)	7 (32%)
	TT	80 (40%)	66 (52%)	51 (49%)	89 (51%)	18 (41%)	60 (50%)	11 (50%)
	A-allele	120 (60%)	62 (49%)	53 (51%)	84 (48%)	26 (59%)	59 (50%)	11 (50%)
NKC9	AA	16 (8%)	19 (15%)	8 (8%)	24 (14%)	1 (2%) ^2^	19 (16%)	5 (23%)
	AG	70 (35%)	30 (23%)	28 (27%)	41 (24%)	16 (36%)	22 (19%)	5 (23%) ^3^
	GG	114 (57%)	79 (62%)	68 (65%)	108 (62%)	27 (61%)	78 (65%)	12 (54%)
	G-allele	184 (92%)	109 (85%)	96 (92%)	149 (86%)	43(97%) ^4^	100(84%)	17(77%) ^5^
NKC10	AA	14 (7%)	5 (5%)	3 (3%)	8 (5%)	1 (2%)	3 (2%)	3 (14%) ^6^
	AT	75 (38%)	44 (34%)	32 (31%)	58 (33%)	15 (34%)	39 (33%)	7 (32%)
	TT	111 (55%)	78 (61%)	69 (66%)	107 (62%)	28 (64%)	77 (65%)	12 (55%)
	T-allele	186 (93%)	122 (95%)	101 (97%)	165 (95%)	43 (98%)	116 (98%)	19 (87%) ^7^
NKC11	CC	80 (40%)	62 (48%)	47 (45%)	85 (49%)	14 (32%) ^8^	54 (45%)	10 (46%)
	CT	91 (46%)	51 (40%)	47 (45%)	69 (40%)	25 (57%)	55 (46%)	7 (32%)
	TT	29 (14%)	15 (12%)	10 (10%)	19 (11%)	5 (11%)	10 (8%)	5 (23%)
	T-allele	120 (60%)	66 (52%)	57 (55%)	88 (51%)	30 (68%) ^9^	65 (54%)	12 (55%)
NKC12	CC	32 (16%)	21 (16%)	14 (13%)	26 (15%)	7 (16%)	14 (12%)	6 (27%)
	CG	87 (44%)	50 (39%)	49 (47%)	70 (40%)	25 (57%)	55 (46%)	7 (32%)
	GG	81 (40%)	57 (45%)	41 (39%)	77 (45%)	12 (27%) ^10^	50 (42%)	9 (41%)
	C-allele	119 (60%)	71 (55%)	63 (60%)	96 (55%)	32 (73%) ^11^	69 (58%)	13 (59%)
NKC17	CC	35 (17%)	23 (18%)	20 (19%)	31 (18%)	9 (21%)	23 (19%)	5 (23%)
	CG	89 (45%)	64 (50%)	47 (45%)	87 (50%)	16 (36%)	55 (46%)	8 (36%)
	GG	76 (38%)	41 (32%)	37 (36%)	55 (32%)	19 (43%)	41 (35%)	9 (41%)
	C-allele	124 (62%)	87 (68%)	67 (64%)	118 (68%)	25 (57%)	78 (65%)	13 (59%)

SLN: sentinel lymph node ^1,7^ SLN metastasis vs. nonmetastasis (*p* = 0.049, Pc = 0.39, OR = 1.32 and *p* = 0.049, Pc = 0.39, OR = 0.16). ^2,8,10^ Ulceration vs. non-ulceration (*p* = 0.033, Pc = 0.79, OR = 0.1; *p* = 0.043, Pc = 1.03, OR = 0.48; *p* = 0.04, Pc = 0.96, OR = 0.46). ^3,6^ SLN metastasis vs. nonmetastasis (*p* = 0.042, Pc = 1.01, OR = 3.4 and *p* = 0.049, Pc = 1.18, OR = 6.1). ^4,9,11^ Ulceration vs. Non-ulceration (*p* = 0.033, Pc = 0.26, OR = 7; *p* = 0.043, Pc = 0.34, OR = 2; *p* = 0.041, Pc = 0.33, OR = 2.1, respectively). ^5^ SLN metastasis vs. controls (*p* = 0.042, Pc = 0.34, OR = 0.3). *p*-value was determined by two-sided Fisher’s exact test.

**Table 4 cancers-11-00438-t004:** *NKG2D* Hb-1 haplotype frequency in healthy controls and melanoma patients.

NKG2D Hb-1 Haplotypes	Controls	Melanoma Patients		
		Total	*p*/Pc	(X^2^)
NK-1	0.587	0.619	0.33	(0.94)
NK-2	0.329	0.276	0.09	(2.87)
NK-3	0.015 ^1^	0.048	0.007/0.098	(7.26)
NK-4	0.010	0.011	0.93	(0.008)
NK-5	0.010	0.011	0.92	(0.009)
NK-6	0.010	0.009	0.82	(0.052)
NK-7	0.005	0.013	0.23	(1.45)
NK-8	0.005	0.007	0.74	(0.11)
NK-9	0.005	0.006	0.84	(0.04)
NK-10	0.010 ^2^	0.000	0.03/0.42	(4.56)
NK-11	0.005	0.000	0.14	(2.19)
NK-12	0.003	0.000	0.27	(1.19)
NK-13	0.003	0.000	0.31	(1.04)
NK-14	0.003	0.000	0.27	(1.21)

Healthy controls and melanoma patients (*n* = 400 and 466 chromosomes respectively). ^1^
*p*/Pc = 0.007/0.098 and X^2^ = 7.26; ^2^
*p*/Pc = 0.03/0.42 and X^2^ = 4.56. *p*-value was determined by Chi-square test.

**Table 5 cancers-11-00438-t005:** Frequency of *NKG2D* Hb-2 haplotypes in healthy controls and melanoma patients.

NKG2D Hb-2 Haplotype	Controls	Melanoma Patients		
		Total	*p*/Pc	(X^2^)
NK-1	0.717	0.732	0.63	(0.23)
NK-2	0.240	0.197	0.13	(2.30)
NK-3	0.000	0.039	0.00009/0.0006	(15.25)
NK-4	0.013	0.024	0.24	(1.35)
NK-5	0.015	0.004	0.10	(2.69)
NK-6	0.012	0.004	0.19	(1.67)
NK-7	0.003	0.000	0.31	(1.04)

Healthy controls and melanoma patients (*N* = 400 and *N* = 466 chromosomes respectively). *p*-value was determined by chi-squared test.

**Table 6 cancers-11-00438-t006:** Frequency of the NK-3 haplotype of the *NKG2D* Hb-2 block in melanoma patients according to histological and clinical characteristics at diagnosis.

NKG2D Hb-2 Haplotype	Melanoma Patients							
	SSM	NM	Tumor Thickness		Ulceration		SLN Metastasis	
			<1 mm	>1 mm	No	Yes	No	Yes
NK-3	0.040	0.034	0.056	0.024	0.046	0.000 ^1^	0.067	0.045

^1^*p*/Pc = 0.040/0.28 and X^2^ = 4.22.

**Table 7 cancers-11-00438-t007:** Association between *NKG2D* haplotypes and genotypes with melanoma.

NKG2D Haplotypes and Genotypes	Controls (*N* = 200)	Melanoma Patients (*N* = 233)		
		Total (*N* = 191)	SSM (*N* = 162)	NM (*N* = 29)
**Hb-1**				
NK-1	163 (81%)	197 (84%)	135 (83%)	26 (90%)
NK-2	109 (54%)	110 (47%)	74 (46%)	16 (55%)
NK-1/NK-1	73 (37%)	92 (40%)	68 (42%)	9 (31%)
NK-1/NK-2	80 (40%)	84 (36%)	55 (34%)	14 (48%)
NK-2/NK-2	24 (12%)	20 (9%)	15 (9%)	2 (7%)
**Hb-2**				
NK-1	182 (91%)	204 (88%)	139 (86%)	26 (90%)
NK-2	84 (42%)	84 (36%)	59 (36%)	11 (38%)
NK-1/NK-1	106 (53%)	137 (59%)	96 (59%)	15 (52%)
NK-1/NK-2	65 (33%)	56 (24%)	37 (23%) ^1^	8 (28%)

^1^ SSM vs. control (*p* = 0.033; OR = 0.601; 95% CI = 0.375–0.962). Statistical significance was examined by the Χ^2^ test. ORs were calculated along with 95% CI values.

**Table 8 cancers-11-00438-t008:** TaqMan assays used for SNP typing of the NKC region.

NKC	SNP ID	AB TaqMan Assay	Nucleotide Variation *
3	rs1049174	C_9345347_10	[G/C]
4	rs2255336	C_22274476_10	[A/G]
7	rs2617160	C_1841959_10	[A/T]
9	rs2246809	C_1842497_10	[A/G]
10	rs2617169	CUSTOM	[T/A]
11	rs2617170	C_1842316_10	[T/C]
12	rs2617171	C_26984346_10	[C/G]
17	rs1983526	C_11919464_10	[C/G]

* https://www.ncbi.nlm.nih.gov/snp.

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
