# Peer review of "NKG2D Polymorphism in Melanoma Patients from Southeastern Spain"

_cancers, 2019, doi:10.3390/cancers11040438_

Round 1

Reviewer 1 Report

This manuscript describes the polymorphisms of NKC region in the southeastern Spain cohort including melanoma patients. In general, the data failed to show clear association between NKC polymorphisms and melanoma. While the data has a worth from epidemiological point of view.

(Major points)

1.  Some values would be wrong. I point out some examples below (not all). I strongly recommend to carefully confirm all values in this manuscript.

Table 2, “MN” in a heading -> “NM”?

Table 2, “5%” in NKC10 T-allele of controls -> “93%”?

Table 2, “99%” in NKC10 T-allele of SSM -> “96%”?

2.  In Fig. 1B, the nucleotide variation of NKC-4 is C/T; however, this variation is given as A/G in table 7 (also in ref. #28). Which is correct?

(Minor points)

3.  In tables, some values are presented in bold font. What does this mean? Perhaps, it means p<0.05. Proper descriptions should be given.  

4.  I think it is interesting that the haplotype of Hb-2 NK-3 was found in only melanoma patients. Therefore, frequencies of this haplotype among melanoma types (histological subtypes and clinical characteristics) should be shown.

5.  The same abbreviations representing NKG2D haplotypes are commonly used in hb-1 and hb-2 (NK-1~NK-7). This is confusing. If these names are given by the authors, it is better to use different abbreviations.

6.  No description about the sample source for the linkage disequilibrium analysis in the materials section.

Author Response

 Major points

1.  Some values would be wrong. I point out some examples below (not all). I strongly recommend to carefully confirm all values in this manuscript.

Table 2, “MN” in a heading -> “NM”?

You are right; the correct abbreviation for Nodular Melanoma is NM. We have carefully reviewed the manuscript to confirm values in tables and text.

Table 2, “5%” in NKC10 T-allele of controls -> “93%”?

Yes, that is correct; NKC10 T-allele is present in 93% of controls.

Table 2, “99%” in NKC10 T-allele of SSM -> “96%”?

Yes, that is correct; NKC10 A-allele is present in 96% of controls.

We have detected more typed errors in Table 2 and 3. They have been highlighted in the text (NKC7 A-allele is present in 49% of total melanoma patients instead of in 50%; the C-allele is present in 66% of total patients instead of 67 and in 72% of NM patients instead of 73, etc.

2.  In Fig. 1B, the nucleotide variation of NKC-4 is C/T; however, this variation is given as A/G in table 7 (also in ref. #28). Which is correct?

Indeed, the correct variation of NKC4 is A/G; there is a mistake in FIG. 1B regarding the nucleotide given for position NKC4.The correct nucleotide in the NK-4 haplotype for hb-2 is A instead of T.

Minor points

3.  In tables, some values are presented in bold font. What does this mean? Perhaps, it means p<0.05. Proper descriptions should be given.  

Yes, bold font meant p<0.05 but since values are indicated in superscript numbers, we have decided to avoid bold fonts.

 4.  I think it is interesting that the haplotype of Hb-2 NK-3 was found in only melanoma patients. Therefore, frequencies of this haplotype among melanoma types (histological subtypes and clinical characteristics) should be shown.

We have included a new table-6 showing the Hb-2 NK-3 haplotype frequencies in according to histological subtypes of melanoma and clinical characteristics of patients. The table has been conveniently described in the results section.

 5.  The same abbreviations representing NKG2D haplotypes are commonly used in hb-1 and hb-2 (NK-1~NK-7). This is confusing. If these names are given by the authors, it is better to use different abbreviations.

We have used exactly the same abbreviations described by Hayashi et al. (ref. 28, in our manuscript) in a series of cancer patients in the Japanese population to facilitate comparison.

 6.  No description about the sample source for the linkage disequilibrium analysis in the materials section.

All patients were included in the LD analysis (N=233).This information is now included in section5.4. NKC region haplotype analysis”.

Reviewer 2 Report

The manuscript is done well. Although some minor corrections are needed to be addressed as stated below:

1.      Abstract- Authors should address the objective of the study in Background section.

2.      Introduction is poorly organized and needs to give a look back considering broad readers.

3.      NKG2A and NKG2D, both genes are randomly described in Introduction section and there is no sequential description present which is very confusing for readers.  Considering current proposed work, it is suggested that authors must re-wright their introduction section.

4.      Authors should also include the main conclusion at the end of Introduction.

5.      Please explain about controls in table 1 below table 1 as note.

6.      Why did authors describe/study only SNPs in NKG2D region and NKG2A promoter?  What about NKG2D promoter and NKG2A gene. Have authors searched for these regions also for SNP mining? What was selection criteria for this study?

7.      Section 4 should be addressed as “Materials and Methods” instead of “Patients and Methods”.

8.      Authors are also suggested to recheck thoroughly manuscript for spelling and grammar. Mistakes at many places have been observed.

Author Response

Comments and Suggestions for Authors

The manuscript is done well. Although some minor corrections are needed to be addressed as stated below:

1. Abstract- Authors should address the objective of the study in Background section.

The objective of the study is now included in the Background section of the Abstract.

2.Introduction is poorly organized and needs to give a look back considering broad readers.

New paragraphs have been included in the introduction for a wider comprehension of the work.

3.      NKG2A and NKG2D, both genes are randomly described in Introduction section and there is no sequential description present which is very confusing for readers.  Considering current proposed work, it is suggested that authors must re-wright their introduction section.

Thank you for your suggestion; we have reorganized the introduction. It should be clearer now.

4.      Authors should also include the main conclusion at the end of Introduction.

We have included the main conclusion at the end of Introduction.

5.      Please explain about controls in table 1 below table 1 as note.

We have included details for controls in a foot note in table 1.

6.      Why did authors describe/study only SNPs in NKG2D region and NKG2A promoter?  What about NKG2D promoter and NKG2A gene. Have authors searched for these regions also for SNP mining? What was selection criteria for this study?

This information is nicely described in the paper of Hayashi et al. (ref. 28 in our manuscript). In that paper, Hayashi et al used a Celera Genomic database to screen marker SNPs in the NKC gene region. Over 1,300 SNPs in the NKC regionhave been registered in the Celera Genomic and NCBI databases. They selected 20 SNPs with allele frequencies >10% in Caucasian and Japanese populations. These 20 SNPs, named NKC-1 to NKC-20, also showed variant allele frequencies >10% in our study population; the SNPs from NKC-1 to NKC-20 cover the CD94, NKG2D, NKG2F, NKG2E, NKG2A, and Ly49 genes. Hayashi et al. found that eight out of those 20 SNPs were significantly associated with natural cytotoxic activity (P <0.001).

Taking this into account, we have indicated in section 5.3 of Materials and Methods that: “Following the criteria described by Hayashi et al. these SNPs were selected among 1,300 SNPs registered in this region in the Celera Genomic and NCBI databases. Briefly, 1) allele frequency >10% in Caucasian and Japanese populations and 2) strong association with NK cell cytotoxic activity (P values <0.001).

7.      Section 4 should be addressed as “Materials and Methods” instead of “Patients and Methods”.

This has been changed.

8.      Authors are also suggested to recheck thoroughly manuscript for spelling and grammar. Mistakes at many places have been observed.

We have reviewed the manuscript for spelling and grammar mistakes.

Reviewer 3 Report

The manuscript by Gimeno et al reports NKG2D polymorphism in melanoma patients from southeastern Spain. Authors are advised to compare the results from other ethnic groups to make their data more meaningful.

Author Response

The manuscript by Gimeno et al reports NKG2D polymorphism in melanoma patients from southeastern Spain. Authors are advised to compare the results from other ethnic groups to make their data more meaningful.

To our knowledge, no other studies have carried out the analysis of polymorphisms in the NKC region in melanoma patients. This fact has been added to the first paragraph of the discussion. Besides, we have re-written that paragraph to compare results for some of the genetic variants analyzed in our study with those obtained from other types of cancer or ethnic groups.

Round 2

Reviewer 3 Report

Authors have responded to my satisfaction.